# Anti-Ageing Potential of *S. euboea* Heldr. Phenolics

**DOI:** 10.3390/molecules26113151

**Published:** 2021-05-25

**Authors:** Ekaterina-Michaela Tomou, Christina D. Papaemmanouil, Dimitrios A. Diamantis, Androniki D. Kostagianni, Paschalina Chatzopoulou, Thomas Mavromoustakos, Andreas G. Tzakos, Helen Skaltsa

**Affiliations:** 1Department of Pharmacognosy & Chemistry of Natural Products, School of Pharmacy, National and Kapodistrian University of Athens, Panepistimiopolis, Zografou, 15771 Athens, Greece; ktomou@pharm.uoa.gr; 2Department of Chemistry, Section of Organic Chemistry and Biochemistry, University of Ioannina, 45110 Ioannina, Greece; christina.pa@hotmail.gr (C.D.P.); dimitrisdiamantis0@gmail.com (D.A.D.); andronikikostagianni@gmail.com (A.D.K.); 3Hellenic Agricultural Organization DEMETER, Institute of Breeding and Plant Genetic Resources, IBPGR, Department of Medicinal and Aromatic Plants, Thermi, 57001 Thessaloniki, Greece; xatzlin@yahoo.gr; 4Department of Chemistry, National and Kapodistrian University of Athens, Panepistimiopolis, 15771 Zografou, Greece; tmavrom@chem.uoa.gr; 5Institute of Materials Science and Computing, University Research Center of Ioannina (URCI), 451100 Ioannina, Greece

**Keywords:** cultivated *Sideritis euboea*, apigenin 7-*O*-p-coumaroylglucosides, lignan, neolignan, *cis*-acteoside, *cis*-leucosceptoside A, 2D-NMR, hyaluronidase, in silico docking, anti-ageing activity

## Abstract

In recent years, the use of *Sideritis* species as bioactive agents is increasing exponentially. The present study aimed to investigate the chemical constituents, as well as the anti-ageing potential of the cultivated *Sideritis euboea* Heldr. The chemical fingerprinting of the ethyl acetate residue of this plant was studied using 1D and 2D-NMR spectra. Isomeric compounds belonging to acylated flavone derivatives and phenylethanoid glycosides were detected in the early stage of the experimental process through 2D-NMR techniques. Overall, thirty-three known compounds were isolated and identified. Some of them are reported for the first time not only in *S. euboea*, but also in genus *Sideritis* *L*. The anti-ageing effect of the ethyl acetate residue and the isolated specialized products was assessed as anti-hyaluronidase activity. In silico docking simulation revealed the interactions of the isolated compounds with hyaluronidase. Furthermore, the in vitro study on the inhibition of hyaluronidase unveiled the potent inhibitory properties of ethyl acetate residue and apigenin 7-*O*-*β*-d-glucopyranoside. Though, the isomers of apigenin 7-*O*-p-coumaroyl-glucosides and also the 4′-methyl-hypolaetin 7-*O*-[6′′′-*O*-acetyl-*β*-d-allopyranosyl]-(1→2)-*β*-d-glucopyranoside exerted moderate hyaluronidase inhibition. This research represents the first study to report on the anti-hyaluronidase activity of *Sideritis* species, confirming its anti-inflammatory, cytotoxic and anti-ageing effects and its importance as an agent for cosmetic formulations as also anticancer potential.

## 1. Introduction

Natural Products (NPs) continue to be the major source for drug leads, providing unique structurally diverse chemical scaffolds [1]. In the chemistry of NPs, Nuclear Magnetic Resonance (NMR) turns out to be an indispensable tool in the process of screening crude plant extracts for bioactive metabolites [2]. However, dereplication and analysis of these extracts still remain a challenging research field for NMR spectroscopy. Plant extracts consist of unique and convoluted mixtures of bioactive NPs with complicated and unpredicted spectral patterns. Over the years, NMR experiments such as two-dimensional (2D-) NMR have enhanced and accelerated the costly and time-consuming analytical process of these extracts, developing new strategies in the design of drug candidates [3].

*Sideritis euboea*, a member of the genus *Sideritis L.* (Lamiaceae family), is an endemic species of Greece, occurring in Evia (Euboea) Island [4]. Traditionally, its infusion (known as mountain tea) is widely consumed against mild gastrointestinal discomfort and common flu [5,6,7,8]. Previous studies reported the rich chemical profile of its extracts, including various metabolites with significant pharmacological effects [4,5,6,7,8,9]. The constantly increasing demand for mountain tea results in serious concerns about the survival of the wild *Sideritis* populations due to their overharvesting. In parallel, the intensifying climate change combined with overharvesting is a particular threat to the biodiversity of these wild plants. Therefore, the cultivation of *Sideritis* species has been viewed as a dynamic solution that could contribute to reducing the extensive collection (or overharvesting) and the potential extinction of the natural wild populations, to produce raw materials in sufficient quantities for the industrial-scale production demanded, and most importantly, to affirm for high quality and standardized products.

As a continuous endeavor to explore and specify the complete phytochemical load of the cultivated species *S. euboea*, we present herein the isolation and identification of 33 chemical constituents from the ethyl acetate residue, including one fatty acid derivative, two diterpenes, three iridoids, twelve flavonoid derivatives, two lignans, six phenylethanoid glycosides and seven phenolic acid derivatives. To evaluate the anti-inflammatory activity of the specific residue of *S. euboea*, as well as its isolated secondary metabolites, an in vitro assay towards the inhibition of the enzyme Hyaluronidase (Hyal) was performed. Hyaluronidase is a glycosidase, whose main role is the degradation of hyaluronic acid, one of the major constituents of the extracellular matrix connective tissue and regulates the homeostasis and the humidity levels of the joints [10,11]. In humans, hyaluronidase can be found both in body fluids, such as sperm or blood, and in various organs, such as skin, eye, uterus, testis, or liver. Except for the degradation of hyaluronic acid, hyaluronidase has been found to take part in various inflammatory pathways, cancer metastasis, and many allergic reactions [12]. The inhibition of hyaluronidase is of great significance, mainly due to its vital role in skin early-ageing as well as for tumour development, as hyaluronidase can be considered as a tumour marker. As a result, skincare cosmetic products with high potency against the activity of hyaluronidase have been well developed. In addition, natural specialized products acting as hyaluronidase inhibitors are lead-compounds to the development of new anti-ageing products [13,14]. The anti-hyaluronidase potential of the isolated compounds was evaluated in silico and several positive hits were determined. The in silico results were further validated in vitro using a hyaluronidase assay and the recorded results were further rationalized based on their in silico determined interaction profile with hyaluronidase.

## 2. Results

### 2.1. Phytochemical Content

The methanol extract of the aerial parts of the cultivated *S. euboea* Heldr. was suspended in boiling water and then, this water-soluble fraction was successively partitioned with ethyl acetate (EtOAc) and *n*-butanol (*n*-BuOH) solvents. 1D and 2D-NMR spectra of the EtOAc residue are shown in Appendix A. The ^1^H-NMR fingerprinting allowed the identification of its major chemical classes such as diterpenes, iridoids, flavonoids, lignans, phenylethanoid glycosides, phenolic acids, and sugars. The δ_H_ ranges of the assigned resonancesfor each class of chemical compounds are presented in Table 1. Furthermore, through 2D-NMR spectra (COSY, HSQC and HMBC) the corresponding signals of these chemical classes were also verified. Notably, we were able to observe olefinic proton signals at δ 5.88–5.70 in the ^1^H-NMR spectrum, which were originally attributed to the protons of the iridoid skeleton. However, the 2D ^1^H–^1^H–COSY spectrum demonstrated correlation peaks between these protons with protons in the aromatic area at δ 6.97–6.88, giving rise to the assumption that these protons belong to *cis*-double bond groups, which are near to phenolic rings (Appendix A). This assumption was firstly confirmed by 2D ^1^H-^13^C HSQC spectrum where the corresponding carbon signals of the protons δ 6.97–6.88 were correlated to δ 144.5–144.7, while the carbon signals of the protons δ 5.88–5.70 were identified at δ 115.0–115.6 (Appendix A). Secondly, the 2D ^1^H-^13^C HMBC experiment revealed cross-peaks between the latter protons and carbons belonging to phenolic rings δ 126.5–128.8 (Appendix A). Based on previous phytochemical reports on genus *Sideritis*, we assumed that the aforesaid assignments could be related to the presence of flavone *cis*-p-coumaroyl glucosides or/and *cis*-phenylethanoid glycosides. As a result, 2D NMR allowed us to identify the core skeletons of complex chemical compounds in the plant mixture, even the isomeric derivatives, through high-resolution data. It is noteworthy to point out that the *cis* derivatives were detected in small amounts since their *trans* isomers which are generally more stable were dominant.

After the spectra interpretation, the EtOAc residue was applied to several chromatographic techniques, including Column Chromatography (CC) over silica gel and Sephadex LH-20, preparative-Thin Layer Chromatography (prep-TLC), to obtain 33 compounds. Subsequently, the present study unveiled one fatty acid derivative: triolein (**1**) [15], two diterpenoids: eubotriol (**2**) [4], siderol (**3**) [16], three iridoids: 8-e*pi*-loganin (**4**) [17], ajugoside (**5**) [18], melittoside (**6**) [19], twelve flavonoid derivatives: kaempferol (**7**) [20], apigenin (**8**) [21], apigenin 7-*O*-β-d-glucopyranoside (**9**) [22], apigenin 7-*O*-[3′′-*O*-*trans*-p-coumaroyl]-β-d-glucopyranoside (**10**) [23], apigenin 7-*O*-[4′′-*O*-*trans*-p-coumaroyl]-*β*-d-glucopyranoside (**11**) [24], apigenin 7-*O*-[4′′-O-*cis*-p-coumaroyl]-*β*-d-glucopyranoside (**12**) [25], apigenin 7-*O*-[6′′-*O*-*trans*-p-coumaroyl]-*β*-d-glucopyranoside (**13**) [24], isoscutellarein 7-*O*-[6′′′-*O*-acetyl-*β*-d-allopyranosyl]-(1→2)-*β*-d-glucopyranoside (**14**) [26], isoscutellarein 7-*O*-[6′′′-*O*-acetyl-*β*-d-allopyranosyl]-(1→2)-6′′-*O*-acetyl-*β*-d-glucopyranoside (**15**) [26], hypolaetin 7-*O*-[6′′′-*O*-acetyl-*β*-d-allopyranosyl]-(1→2)-*β*-d-glucopyranoside (**16**) [26], 4′-methyl-hypolaetin 7-*O*-[6′′′-*O*-acetyl-*β*-d-allopyranosyl]-(1→2)-*β*-d-glucopyranoside (**17**) [26], 4′-methyl-hypolaetin 7-*O*-[6′′′-*O*-acetyl-*β*-d-allopyranosyl]-(1→2)-6′′-*O*-acetyl-*β*-d-glucopyranoside (**18**) [26], two lignans: pinoresinol-4-*O*-*β*-d-glucopyranoside (**19**) [27], (7*S*, 8*R*)-urolignoside (**20**) [28], six phenylethanoid glycosides: *trans*-acteoside (**21**) [29], *cis*-acteoside (**22**) [30], *trans*-leucosceptoside A (**23**) [31], *cis*-leucosceptoside A (**24**) [32], martynoside (**25**) [33], lamalboside (**26**) [30] and seven phenolic acid derivatives: p-coumaric acid (**27**) [34], methyl *trans* p-coumarate (**28**) [35], 4-methoxybenzoic acid (**29**) [36], 4-hydroxybenzoic acid (**30**) [37], vanillic acid (**31**) [38], syringic acid (**32**) [39], benzyl-1-*O*-*β*-d-glucopyranoside (**33**) [40] (Appendix A). The chemical structures of all the isolated compounds were determined based on the interpretation of spectroscopic data (1D and 2D NMR), as well as on previous literature data.

Compounds **1**–**3** were isolated from the non-polar fractions of the plant residue. This study represents the first report of triolein in genus *Sideritis*, whereas compounds **2** and **3** were extensively found in this genus and might also be characteristic compounds for the specific species [4,9,41]. Furthermore, the three isolated iridoids (compounds **4**–**6**) were previously reported on genus *Sideritis* [8,42,43,44]. Though, 8-*epi*-loganin (**4**) and ajugoside (**5**) were not isolated from *S. euboea*, up to now.

Genus *Sideritis* is very rich in flavonoid derivatives [45,46,47]. In this study, twelve flavonoid analogues were isolated which were categorized into one 3-hydroxyflavone (**7**), one flavone (**8**), one flavone 7-*O*-glucoside (**9**), four flavone 7-*O*-p-coumaroyl-glucosides (**10**–**13**) and five flavone 7-*O*-allosylglucosides (**14**–**18**). Tsaknis and Lalas (2004) isolated the 3-hydroxyflavone, namely kaempferol (**7**) from the wild *S. euboea*, also reporting its great antioxidant activity [5]. Our results confirmed the presence of kaempferol in *S. euboea*. Compounds **8** and **9** were mentioned in the specific species in the precedent study [8]. Regarding the isolation and the identification of the apigenin 7-*O*-p-coumaroyl-glucosides, 1D and 2D- NMR methods played a determinant role, enhancing the detection of each derivative and aiding in carefully identifying their assignments. In the NMR spectra of compounds **10** and **11**, a distinct difference was observed at the signals of their glucose moieties. Specifically, the proton H-3′′ of apigenin 7-*O*-[3′′-*O*-*trans*-p-coumaroyl]-*β*-d-glucopyranoside (**10**) was spotted at δ 5.16 (δc 78.6, HSQC), while the proton H-4′′ of apigenin 7-*O*-[4′′-*O*-*trans*-p-coumaroyl]-β-d-glucopyranoside (**11**) was observed at 4.98 (δc 71.5, HSQC). After fractionations, we succeeded to isolate pure compound **11**, while the separation and isolation of its *cis*-isomer, compound **12**, was not feasible. However, we tracked down the proton signals of these two isomeric acylated flavones, using ^1^H-NMR, COSY, NOESY, HSQC and HMBC spectra. Regarding the NMR spectra of compounds **11** and **12**, we assumed the following important observations for the p-coumaroyl group which could be used as characteristic markers; i) the olefinic protons of the *cis*-isomer are significantly shielded at δ 6.94, d 11.9 Hz (δc 146.0, HSQC) and δ 5.86, d 11.9 Hz (δc 115.8, HSQC) compared to the protons of the *trans*-isomer at δ 7.70, d 15.2 Hz (δc 146.5, HSQC) and δ 6.44, d 15.2 Hz (δc 115.0, HSQC) and ii) the aromatic protons of *cis*-isomer H-2′′′/6′′′ were strongly deshielded at δ 7.71 (d, 8.7 Hz, H-2′′′/6′′′, δc 133.7 HSQC), while the protons H-3′′′/5′′′ were found upfield at δ 6.77 (d, 8.7 Hz, δc 115.5 HSQC). It is important to stress out that we also observed the NOE signal between the olefinic protons of the *cis*-isomer in the NOESY spectrum, confirming the presence of the *cis*-derivative.

Comparing the ^1^H-NMR profiling of apigenin 3′′-/ 4′′-p-coumaroyl-glucosides (**10**–**12**) to apigenin 6′′-p-coumaroyl-glucoside (**13**), we detected two characteristic differences. Firstly, the methylene protons of glucose skeletons of compound **13** were strongly deshielded (δ 4.63, dd 11.9/2.3 Hz, H-6′′a and 4.29, dd 11.9/8.3 Hz, H-6′′b), as well as it was also observed a large upfield shift in the aromatic protons of aglycon and of p-coumaroyl moiety of the latter compound. According to literature, apigenin 4′′ and 6′′-p-coumaroyl-glucosides were previously reported in several *Sideritis* species [24,45,48]. Of great interest is that the majority of these studies do not mention the isomer of apigenin 4′′-p-coumaroyl-glucoside, whereas few of them describe the isolation of the *trans*-acylated flavone glucoside. To the best of our knowledge, the presence of apigenin 7-*O*-p-coumaroyl-glucosides (**10**–**13**) is mentioned for the first time in *S. euboea*. Though, apigenin 7-*O*-[3′′-*O*-*trans*-p-coumaroyl]-*β*-d-glucopyranoside (**10**) is found for the second time in genus *Sideritis* [49]. The ^1^H-NMR spectra of all the isolated apigenin derivatives are presented in Figure 1.

Lignans and neolignans consist of a large group of naturally occurring phenols in the Lamiaceae family. Nonetheless, these compounds are of rare occurrence in genus *Sideritis*. A current study carried out by Kirmizibekmez et al. (2019) mentioned the isolation of two lignan glucosides from the aerial parts of the wild *S. germanicopolitana*, namely dehydrodiconiferylalcohol 4-*O*-*β*-d-glucopyranose and pinoresinol 4′-*O*-*β*-glucopyranoside [50]. Our research revealed the isolation and identification of two lignan analogues, a furofuran skeleton derivative namely pinoresinol-4-*O*-β-d-glucopyranoside (**19**) and a neolignan, known as (7*S*, 8*R*)-urolignoside (**20**). These specialized products are found for the first time in this species. Notably, this is the first report of (7*S*, 8*R*)-urolignoside in genus *Sideritis*.

Phenylethanoid glycosides (PhGs) are among the major compounds found in genus *Sideritis* [47]. Several structures of this chemical category were thoroughly described in various *Sideritis* species, including mainly PhGs with two or three sugars. Compounds **21**, **23** and **25** were previously found in *S. euboea* [6,7,8], while lamalboside (**26**) is reported in the specific species for the first time, herein. Furthermore, *cis*-acteoside (**22**) and *cis*-leucosceptoside A (**24**) are new for the genus *Sideritis*. It is noteworthy to point out that compound **22** was isolated as a mixture with its isomer **21**. The NMR techniques enabled us to obtain all the necessary information to identify these specialized products, even if the *cis* analogue was observed in a minor amount and only in one sample. As a first step, in the ^1^H-NMR spectrum, we identified the proton signals of compound **21** and then, we noticed the rest proton peaks of compound **22**. It was observed that the two isomers were mainly differentiated in two major points in 1D-/2D-NMR spectra; i) the proton signals of the olefinic protons of the caffeoyl moiety, i.e., δ 6.88 (d, 12.1 Hz, H-7′′′/ δc 147.3, HSQC) and δ 5.78 (d, 12.1 Hz, H-8′′′/ δc 115.8, HSQC) for the *cis*-isomer and ii) the proton signals of the aromatic protons of the caffeoyl moiety, i.e., δ 7.53 (d, 2.2 Hz, H-2′′′/ δc 118.6, HSQC), δ 7.11 (dd, 8.5/2.2 Hz, H-6′′′/ δc 125.4, HSQC) and δ 6.75 (d, 8.5 Hz, H-5′′′/ δc 115.3, HSQC) for the *cis*-isomer. As a result, it is important to underlie that the large downfield of proton H-2′′′, as well as the upfield of protons H-7′′′ and H-8′′′ among with their multiplicity (d~ 11–12 Hz) in 1D-NMR might be used as characteristic markers for the presence of the *cis*-acteoside (Figure 2). Moreover, our findings unveiled the appearance of one more additional minor *cis*-isomer specialized product, namely *cis*-leucosceptoside A (**24**). In 1D NMR fingerprinting, we discerned proton signals corresponding to the protons of a mixture of compounds **23** and **24** (Figure 3). The latter isomers were differentiated in the proton signals of their feruloyl moieties in ^1^H-NMR spectrum, i.e., δ 7.88 (d, 2.0 Hz, H-2′′′), δ 7.17 (dd, 8.5/2.0 Hz, H-6′′′), δ 6.94 (d, 11.7 Hz, H-7′′′), δ 6.77 (d, 8.5 Hz, H-5′′′), δ 5.80 (d, 11.7 Hz, H-8′′′) for compound **24** (Figure 3). Notably, we should report the significant downfield of H-2′′′ of the feruloyl group of isomer **24**.

Concerning the isolated phenolic derivatives, compounds **27** and **30**–**33** were reported in previous studies on various *Sideritis* species [49,51,52], whereas compounds **28** and **29** were not found before in the specific genus. However, the constituents **28**–**33** were isolated for the first time in *S. euboea*.

We should also point out that spotting the major chemical categories of the crude extract at the beginning of our workflow enabled us to confirm most of the isolated compounds and minimize potential artifacts during the analytical process.

### 2.2. In Silico Screening and In Vitro Evaluation of the Inhibition of S. euboea Ethyl Acetate Residue and Its Isolated Compounds towards Hyaluronidase

To explore the anti-hyaluronidase potential of the isolated compounds, we performed inverse virtual screening against hyaluronidase. The calculated free energy of binding of hyaluronidase against each isolated compound is presented in Figure 4. From the plot illustrated in Figure 4, it became evident that apigenin 7-*O*-*β*-d-glucopyranoside (compound **9**) and its acylated derivatives (compounds **10**–**13**) showed the most optimum binding profile with free binding energy values of −10.4, −10.4, −10.4, −10.5 and −10.8 kcal/mol, respectively. For the diterpenoids eubotriol (compound **2**) and siderol (compound **3**), the calculated free energy of binding was determined as −8.8 kcal/mol and −7.0 kcal/mol, respectively. The iridoids 8*-epi*- loganin (compound **4**), ajugoside (compound 5) and melittoside (compound **6**) determined to have free energy binding with values of −6.9 kcal/mol, −7.0 kcal/mol and −7.3 kcal/mol, respectively. Regarding the rest isolated flavonoids: kaempferol (compound **7**), apigenin (compound **8**), isoscutellarein 7-*O*-[6′′′-*O*-acetyl-*β*-d-allopyranosyl]-(1→2)-*β*-d-glucopyranoside (compound **14**), isoscutellarein 7-*O*-[6′′′-*O*-acetyl-*β*-d-allopyranosyl]-(1→2)-6′′-*O*-acetyl-*β*-d-glucopyranoside (compound **15**), hypolaetin 7-*O*-[6′′′-*O*-acetyl-*β*-d-allopyranosyl]-(1→2)-*β*-d-glucopyranoside (compound **16**), 4′-methyl-hypolaetin 7-*O*-[6′′′-*O*-acetyl-*β*-d-allopyranosyl]-(1→2)-*β*-d-glucopyranoside (compound **17**), 4′-methyl-hypolaetin 7-*O*-[6′′′-*O*-acetyl-*β*-d-allopyranosyl]-(1→2)-6′′-*O*-acetyl-*β*-d-glucopyranoside (compound **18**), the results showed moderate free energy binding values of −7.9 kcal/mol, −7.9 kcal/mol, −8.10 kcal/mol, −8.30 kcal/mol, −7.8 kcal/mol, −8.4 kcal/mol and −8.2 kcal/mol, respectively. In addition, the two lignans pinoresinol-4-*O*-*β*-d-glucopyranoside (compound **19**) and (7*S*, 8*R*)-urolignoside (compound **20**) were determined to bind to hyaluronidase with free energy binding values of −7.6 kcal/mol and −7.4 kcal/mol, respectively. The six phenylethanoid glycosides: *trans*-acteoside (compound **21**), *cis*-acteoside (compound **22**), *trans*-leucosceptoside A (compound **23**), *cis*-leucosceptoside A (compound **24**), martynoside (compound **25**), lamalboside (compound **26**) interacted with hyaluronidase with free energy binding values of −8.4 kcal/mol, −8.7 kcal/mol, −8.5 kcal/mol, −8.3 kcal/mol, −8.0 kcal/mol and −8.4 kcal/mol, respectively. The last family of the following isolated phenolic acid derivatives: p-coumaric acid (compound **27**), methyl *trans* p-coumarate (compound **28**), 4-methoxybenzoic acid (compound **29**), 4-hydroxybenzoic acid (compound **30**), vanillic acid (compound **31**), syringic acid (compound **32**) and benzyl-1-*O*-*β*-d-glucopyranoside (compound **33**) displayed the following free energy binding values −6.1 kcal/mol, −9.4 kcal/mol, −5.6 kcal/mol, −5.7 kcal/mol, −5.6 kcal/mol, −5.7 kcal/mol and −7.2 kcal/mol, respectively.

We then selected compounds **9**, **12**, **13**, which showed enhanced in silico binding affinities to hyaluronidase, as well as compounds **17** and **18** which showed moderate binding affinity, to evaluate in vitro their anti-hyaluronidase activity. The five selected compounds (**9**, **12**, **13**, **17** and **18**) as also the EtOAc residue of *S. euboea* were evaluated through an in vitro screening assay in two concentrations, 300 μg/mL and 500 μg/mL, in order to evaluate the inhibitory activity of each compound at these two concentrations (Figure 5). The EtOAc crude extract of *S. euboea* showed weak inhibition at a concentration of 300 μg/mL, with a value of 18.55 ± 0.12%, whereas at 500 μg/mL, it showed moderate inhibition with a value of 35.67 ± 0.15%. However, compound **9**, which is a derivative of apigenin glucoside illustrated moderate inhibitory activity (34.84 ± 0.08%) at a concentration of 300 μg/mL and very good (64.77 ± 0.06%) inhibitory activity at a concentration of 500 μg/mL. Compounds **12** and **13**, which also consist of derivatives of apigenin glucoside showed weaker inhibition at 300 μg/mL, with values of 11.77 ± 0.06% and 0.8 ± 0.15%, respectively. However, at a concentration of 500 μg/mL compound **12** inhibited hyaluronidase with a value of 34.17 ± 0.08%. Between the two hypolaetin 7-*O*-allosylglucoside derivatives, which were studied for their anti-hyaluronidase potency, compound **17** showed weak inhibitory activity with 7.26 ± 0.11% at a concentration of 300 μg/mL, while at 500 μg/mL, the inhibition was moderate with a value of 35.67 ± 0.06%. Compound **18**, however, showed weak inhibition at a concentration of 500 μg/mL, with a value of 10.96 ± 0.21% (Figure 5). Additional studies were performed to determine the IC_50_ values for the EtOAc residue as well as of the isolated compounds. For the EtOAc residue the IC_50_ values were determined to 690.87 ± 2.01 μg/mL, while for compounds **9**, **12** and **17** the IC_50_ values were 927.15 ± 5.30 μΜ (400.90 ± 0.012 μg/mL), 1170 ± 8.72 μΜ (680 ± 0.015 μg/mL) and 978.01 ± 5.63 μΜ (650 ± 0.084 μg/mL) respectively, while for the compounds **13** and **18**, the IC_50_ values were found to be out of range.

### 2.3. In Silico Docking of Isolated Compounds towards Hyaluronidase

In order to further explore the identified differences in the in vitro activity of the different compounds against hyaluronidase, we analyzed the interaction profile of each compound which was selected via in silico docking, with hyaluronidase. Figure 6 and Appendix A of the Appendix A illustrate the interactions recorded between the five isolated compounds with hyaluronidase.

In Figure 6, it is shown the interaction profile of compound 9 (illustrated in light blue sticks) with hyaluronidase, where numerous hydrophobic, hydrogen bonds and pi-stacking interactions stabilize the formation of a complex. In this binding pose, the sugar moiety of compound **9** interacts with hyaluronidase through hydrophobic bonds with the active site residue Tyr202. Furthermore, the sugar moiety of compound **9** forms hydrogen bonds with the active site residues Glu131, Tyr202 and Asp292 (Figure 6). The strong interaction is also affected through pi-stacking interactions between the flavonoid moiety (apigenin) of compound **9** and the enzyme through the active site aromatic residues Trp321 and Tyr75. In addition, the apigenin skeleton develops hydrophobic interactions with Tyr75, Tyr202, Tyr247, Tyr286 and Trp321. These data illustrate that this molecule strongly interacts specifically with the amino acid residues of the active site of hyaluronidase. These docking results could possibly explain the high inhibitory potency of compound **9** towards hyaluronidase. Compound **12** (Appendix A) forms hydrophobic interactions between its flavonoid moiety and hyaluronidase, through enzyme residues Tyr202, Tyr286, and Trp321, while additional hydrogen bonds are formed between apigenin and the residues Asn37 and Tyr286. Regarding its sugar moiety, it develops interactions with hyaluronidase through hydrogen bonds with Glu131. In addition, the coumaroyl moiety forms hydrogen bonds with Tyr202 as also hydrophobic interactions with Tyr202 and Tyr208. The coumaroyl moiety further forms hydrophobic interactions with Tyr202, Phe204 and Tyr208, while pi-stacking interactions are developed with Phe204. The calculated binding affinity of compound **12** for hyaluronidase was −10.5 kcal/mol. However, the respective inhibition at 300 μg/mL and 500 μg/mL is lower than the respective values of compound **9**, as compound **9** interacts strongly with more active sited amino acid residues than compound **12**.

The glucosidic analogue **13** (Appendix A) develops with hyaluronidase hydrophobic interactions between its coumaroyl moiety and the amino acids Tyr75, Tyr202, Tyr247, Tyr286 and Trp321. The binding of compound **13** with hyaluronidase is stabilized through pi-stacking interactions with the residues Tyr202, and Trp321 also via the coumaroyl moiety, leading to a binding affinity of −10.8 kcal/mol. Regarding the flavonoid moiety, it forms hydrophobic bonds with Tyr75 and Trp321. Lastly, the sugar moiety develops hydrophobic interactions with Tyr247 of hyaluronidase 47. Despite the fact that compound **13** has the most promising binding affinity comparing to the other isolated compounds, compound **9** displays the highest inhibitory potency towards hyaluronidase. This fact could be explained via in silico studies, as the coumaroyl moiety, in contrast to the sugar moiety of compound **13** interacts strongly with the active sited amino acid residues as it was analyzed above.

In contrast, compound **17** (Appendix A) showed a lower binding affinity with a value of −8.1 kcal/mol. In this case, the interaction takes place through hydrogen bonds between the sugar moiety and Arg134, and through hydrophobic bonds with Tyr202. Simultaneously, the flavonoid moiety forms hydrogen bonds with the amino acid residues Glu131, Tyr202, Asp292 and Tyr247. In addition, pi-stacking bond with Trp321 stabilizes the binding of compound **17** to hyaluronidase. The hypolaetin moiety of compound **17**, also displays hydrophobic interactions with the active site amino acid residues Tyr75, Tyr202, Tyr247, and Trp321. In addition, the acetyl-*β*-d-allopyranosyl moiety, without the second sugar residue, of compound **17** interacts with hyaluronidase through hydrophobic bonds with the amino acid residues Phe212, Pro249 and Val251 which, though, do not belong to the active center. As illustrated in Figure 5, compound **17** displays lower inhibitory potency compared to compound **9**. Compared to compound **12**, although their inhibition values are in the same range, they differ in their binding affinities. This fact could be explained due to the formation of hydrophobic interactions between both the coumaroyl and the flavonoid moiety with the active site amino acid residues, whereas in compound **17** only the flavonoid moiety interacts strongly with the amino acids of the active center. Similar binding affinity showed compound **18** (Appendix A), with a value of −8.3 kcal/mol. However, compound **18** shows no inhibitory potency towards hyaluronidase. According to in silico docking results, compound **18** forms no hydrophobic interactions but displays hydrogen bonds with the residues Arg134 and Asp292 which are not part of the active center. This could deliver a potential explanation of the low inhibitory potential of this compound with respect to compound **9**.

## 3. Materials and Methods

### 3.1. General Procedures

1D and 2D NMR spectra were recorded in CD_3_OD and CDCl_3_ on Bruker DRX 400 instrument at 295 K. Chemical shifts are given in ppm (δ) and were referenced to the solvent signals at 3.31/49.0 and 7.24/77.0 ppm, respectively. COSY (COrrelation SpectroscopΥ), HSQC (Heteronuclear Single Quantum Correlation), HMBC (Heteronuclear Multiple Bond Correlation), and NOESY (Nuclear Overhauser Effect SpectroscopYy) (mixing time 950 ms) experiments were performed using standard Bruker microprograms. Column chromatography (CC): Sephadex LH-20 (Pharmacia) and Silica gel (Merck, Art. 9385, Darmstadt, Germany). Preparative–thin-layer chromatography (Prep-TLC) plates were pre-coated with silica gel (Merck, Art. 5721, Darmstadt, Germany). Fractionation was always monitored by TLC silica gel 60 F– 254 (Merck, Art. 5554, Darmstadt, Germany) with visualization under UV (254 and 366 nm) and spraying with vanillin-sulfuric acid reagent. All obtained extracts, fractions, and isolated compounds were evaporated to dryness in a vacuum under low temperature and then were put in activated desiccators with P_2_O_5_ until their weights had stabilized.

### 3.2. Plant Material

Aerial parts of *S. euboea* Heldr. were collected from a cultivated population (under organic farming conditions) in HAO DEMETER (Institute of Breeding and Plant Genetic Resources) in July 2017. The sample was authenticated by Dr. P. Chatzopoulou; a voucher specimen was deposited in the herbarium of the Aromatic and Medicinal Plant Department (code 19–17). The plant material was dried for 10 days at room temperature, and then powdered in a specific pulverization machine without freezing.

### 3.3. Extraction and Isolation

The whole process of the extraction of the air–dried powdered plant material (0.30 kg) was conducted as previously described [8]. The ethyl acetate residue (1.0 g) was fractionated by CC over silica gel (14.0 cm × 3.0 cm), using as eluent mixtures of increasing polarity (dichloromethane:methanol:water) to yield finally 30 fractions (A–Z5). Fractions A and B (30.0 mg; eluted with DCM:MeOH:H_2_O 9.9:0.1:0.01–9.6:0.4:0.04) were combined and were submitted to CC over silica gel (DCM:MeOH:H_2_O 1:0:0 to 0:1:0) and yielded compounds **28** (5.2 mg), **29** (2.8 mg) and **30** (1.0 mg). Fraction E (36.8 mg; eluted with DCM:MeOH:H_2_O 9.4:0.6:0.06) was submitted to CC over silica gel and afforded compounds **1** (3.0 mg), **2** (1.8 mg) and **31** (2.1 mg). Fraction F (8.1 mg) was identified as compound 3. Fractions G (2.3 mg) and H (7.9 mg) (both eluted with DCM:MeOH:H_2_O 9.0:1.0:0.1) were identified as compounds **27** and **8**, respectively. Fraction K (30 mg; DCM:MeOH:H_2_O 9.0:1.0:0.1) was subjected to CC over silica gel and yielded compound 32 (2.5 mg). Fraction O (26.1 mg; eluted with DCM:MeOH:H_2_O 8.5:1.5:0.15) was identified as a mixture of compounds **10** and **11**. Combined fractions Q and R (61.1 mg; eluted with DCM:MeOH:H_2_O 8.0:2.0:0.2) were further subjected to CC over silica gel (DCM:MeOH:H_2_O 1:0:0 to 0:1:0) and yielded **12** subfractions (Q′A–Q′M). Subfraction Q′B (1.0 mg; eluted with DCM:MeOH:H_2_O 8.5:1.5:0.15) was identified as a mixture of the two isomers **11** and **12**. Combined subfractions Q′D and Q′E (10.0 mg; eluted with DCM:MeOH:H_2_O 8.5:1.5:0.15) were purified by prep-TLC on silica gel with EtOAc:MeOH:H_2_O 9:1.5:1.0 and afforded compounds **19** (1.9 mg; Rf:0.49), **33** (2.9 mg; Rf:0.51) and **11** (1.2 mg; Rf:0.69). Combined subfractions Q′F and Q′G (8.0 mg; eluted with DCM:MeOH:H_2_O 8.0:2.0:0.2) were further submitted to prep-TLC over silica gel with EtOAc:MeOH:H_2_O 9:1.5:1.0 and gave compounds **18** (1.2 mg; Rf:0.44), **4** (1.0 mg; Rf:0.53) and **13** (1.0 mg; Rf:0.7). Combined subfractions Q′I and Q′K (12.0 mg; eluted with DCM:MeOH:H_2_O 8.0:2.0:0.2) were further purified to prep-TLC over silica gel with EtOAc:MeOH:H_2_O 9.0:1.5:1.0 and afforded compounds **15** (1.9 mg; Rf:0.46) and **25** (1.0 mg; Rf:0.49). Fraction U (80.3 mg; eluted with DCM:MeOH:H_2_O 8.0:2.0:0.2) was subjected to CC over silica gel with EtOAc:MeOH:H_2_O 1:0:0 to 0:1:0 and yielded compounds **9** (1.4 mg), **17** (1.3 mg), **5** (2.0 mg) and **6** (5.0 mg). Subfraction UC was further purified by prep-TLC on silica gel with EtOAc:MeOH:H_2_O 8.0:1.5:1.0 and gave **18** (2.5 mg; Rf:0.29). Fraction V (58.3 mg; eluted with DCM:MeOH:H_2_O 8.0:2.0:0.2) was subjected to CC over Sephadex using as eluent MeOH (100%) and afforded compounds **20** (1.2 mg) and **7** (1.0 mg). Subfraction VE (7.2 mg) was further purified by prep-TLC using EtOAc:MeOH:H_2_O 8.0:1.5:1.0, affording a mixture of isomers **23** and **24** (1.2 mg; Rf:0.55). Subfraction VH (15.0 mg) was subjected to prep-TLC using EtOAc:MeOH:H_2_O 8.0:1.5:1.0 and afforded compounds **17** (2.6; Rf:0.41), **14** (2.5 mg; Rf:0.50), **9** (1.2 mg; Rf:0.61). Fraction X (13.5 mg; eluted with DCM:MeOH:H_2_O 8.0:2.0:0.2) was submitted to prep-TLC over silica gel with EtOAc:MeOH:H_2_O 8.0:1.5:1.0 and gave compound **14** (2.3 mg; Rf:0.36). Fractions Y and Z (50.0 mg; eluted with DCM:MeOH:H_2_O 8.0:2.0:0.2) were combined and further fractionated by CC over Sephadex using as eluent MeOH (100%) and afforded compounds **23** (1.7 mg) and **14** (4.4 mg). Combined fraction Z′1 and Z′2 (156.2 mg; eluted with DCM:MeOH:H_2_O 7.5:2.5:0.25) were submitted to CC over Sephadex using as eluent MeOH (98%) and yielded a mixture of compounds **21** and **23** (4.6 mg), **14** (4.2 mg) and **16** (1.0 mg). Fraction Z′3 was identified as compound **21** (13.2 mg). Combined fraction Z′4 and Z′5 (86.3 mg; eluted with DCM:MeOH:H_2_O 7.0:3.0:0.3) were submitted to CC over Sephadex using as eluent MeOH (95%) and gave a mixture of compounds **21** and **22** (4.2 mg), **21** (1.1 mg), **26** (1.0 mg), **14** (3.3 mg) and **16** (1.5 mg).

### 3.4. Evaluation of the Inhibition of S. euboea Ethyl Acetate Residue and Its Isolated Compounds towards Hyaluronidase

The activity of hyaluronidase was determined through UV spectroscopy, by measuring the amount of N-acetylglucosaminoglycan, which is formed by the degradation of hyaluronic acid [53]. An amount of 400 U/mL of Hyalurorindase from bovine testes (Type I-S, lyophilized powder, 400–1000 units/mg, Sigma, Germany) was dissolved in 0.1 M acetate buffer, pH 3.5 and incubated with the aqueous solutions of the respective ligands at concentrations of 300 μg/mL and 500 μg/mL for 20 min at 37 °C. After the incubation of the mixture, 0.5 mg/mL BSA (Bovine Serum Albumin, lyophilized powder, crystallized ≥ 98%, Sigma, Germany) was added in each sample, and then water to reach a total volume of 200 μL. The total mixture was incubated in 37 °C for 60 min. The respective control samples had water in place of the samples. After the reaction time, 148 μL of each sample was transferred to an Eppendorf and 32.8 μL of boric acid was added. The total mixture was incubated at 100 °C for 5 min, to stop the reaction. The samples remained at room temperature (25 °C) and 820 μL of p-dimethylaminobenzaldeyde (DMAB) (cas No:100-10-7, Sigma, Germany) were added. The mixture was incubated at 37 °C for 20 min until the color of the producing N-acetylglucosaminoglycan was developed. The samples are measured at 590 nm by a UV spectrometer.

### 3.5. In Silico Docking of Isolated Compounds towards Hyaluronidase and in Silico Based Screening

For the ligand-based virtual screening, a 3D similarity search is conducted with the software WEGA [54] and the software SHAFTS [55]. For the docking calculations, we utilized the software Autodock Vina after setting a grid of 25 Å^3^ centered in the defined binding site of every protein target. Protein targets and ligands are set for docking by means of AutoDock tools as in Forli et al. (2016) [56].

To validate the docking results we performed docking calculations of the well-known inhibitor of hyaluronidase liquiritigenin that its interaction with hyaluronidase has been reported [57]. For this, we used a different docking software (Maestro). Both software (Maestro and Autodock) resulted in the same interaction profile of liquiritigenin with hyaluronidase (Appendix A). Having determined the validity of our docking protocol, we performed docking calculations of compounds **9, 12, 13, 17** and **18** with Maestro software. The structures of the five selected compounds (**9, 12, 13, 17** and **18**) were built in the Maestro 10.2 utility, prepared with LigPrep, ionized at pH 7.0, minimized using the OPLS3 force field [58], while ConfGen in the thorough mode was used to generate possible bioactive conformations [59]. The crystal structure of hyaluronidase (PDB ID: 2PE4) was prepared using the Protein Preparation Wizard utility [60]. The receptor grid was generated centroid the catalytic site of the enzyme. The GlideXP algorithm was used for the docking of the ligands [61]. All the created poses were minimized post-docking (Figure 6, Appendix A).

## 4. Conclusions

It is well known that plant extracts consist of a large variety of bioactive NPs. However, the chemical composition of the crude extracts is usually very complex. In the drug design process, profiling methods such as NMR techniques play an important role in their investigation. Using NMR methods from the early stages of a phytochemical analysis allows the rapid, non-destructive, reproducible, and costless detection of the major chemical groups of the specialized products, as well as the identification of the unknown or rare compounds.

The present study clearly showed the strength of NMR spectroscopy to identify chemical groups and spot isomeric compounds in plant mixtures, revealing the isolation and identification of many chemical compounds for the first time not only in the species *S. euboea*, but also in the genus *Sideritis*. These data are also of great chemophenetic interest. In vitro study on the inhibition of hyaluronidase by the EtOAc residue of *S. euboea*, as well as the isolated compounds, showed potent inhibitory properties of the residue and apigenin 7-*O*-*β*-d-glucopyranoside (compound **9**) mainly, at a concentration of 500 μg/mL, while the rest apigenin 7-*O*-p-coumaroyl-glucosides and 4′-methyl-hypolaetin 7-*O*-[6′′′-*O*-acetyl-*β*-d-allopyranosyl]-(1→2)-*β*-d-glucopyranoside (compound **17**) showed moderate inhibition at the same concentration. Furthermore, the in silico studies described the kind of interactions of the studied compounds with hyaluronidase. Based on the in silico virtual screening, compounds **9**–**13** showed similar binding affinity, however, their inhibition potential showed different values. In silico docking studies were performed in order to evaluate and explain the interaction profile of each selected compound.

## Figures and Tables

**Figure 1 molecules-26-03151-f001:**
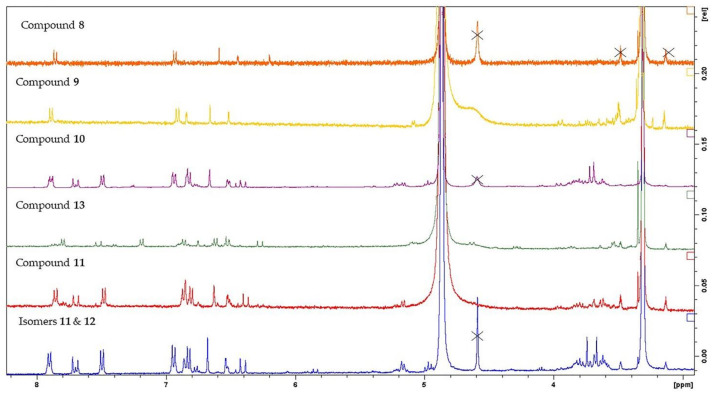
^1^H-NMR spectra of the isolated apigenin derivatives (compounds **8**–**13**).

**Figure 2 molecules-26-03151-f002:**
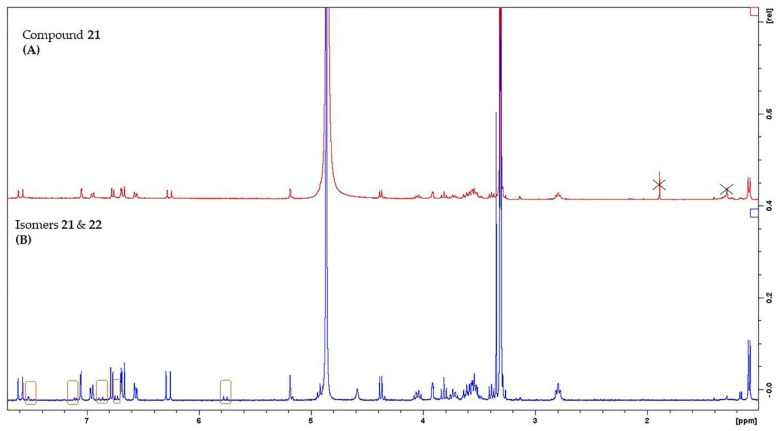
^1^H-NMR spectra of acteoside isomers. (**A**) ^1^H-NMR spectrum of *trans*-acteoside (compound **21**), (**B**) ^1^H-NMR spectrum of the mixture of *trans*-acteoside (compound **21**) and *cis*-acteoside (compound **22**). The major points of differences of the two isomers were signed in orange boxes.

**Figure 3 molecules-26-03151-f003:**
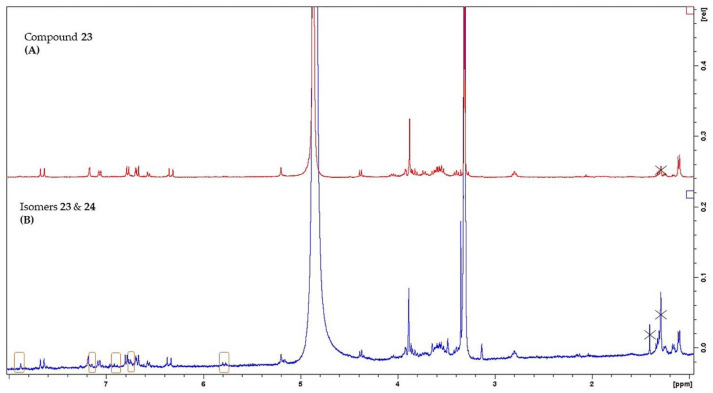
^1^H-NMR spectra of leucosceptoside A isomers. (**A**) ^1^H-NMR spectrum of *trans*-leucosceptoside A (compound **23**), (**B**) ^1^H-NMR spectrum of the mixture of *trans*-leucosceptoside A (compound **23**) and *cis*- leucosceptoside A (compound **24**). The major points of differences of the two isomers were signed in orange boxes.

**Figure 4 molecules-26-03151-f004:**
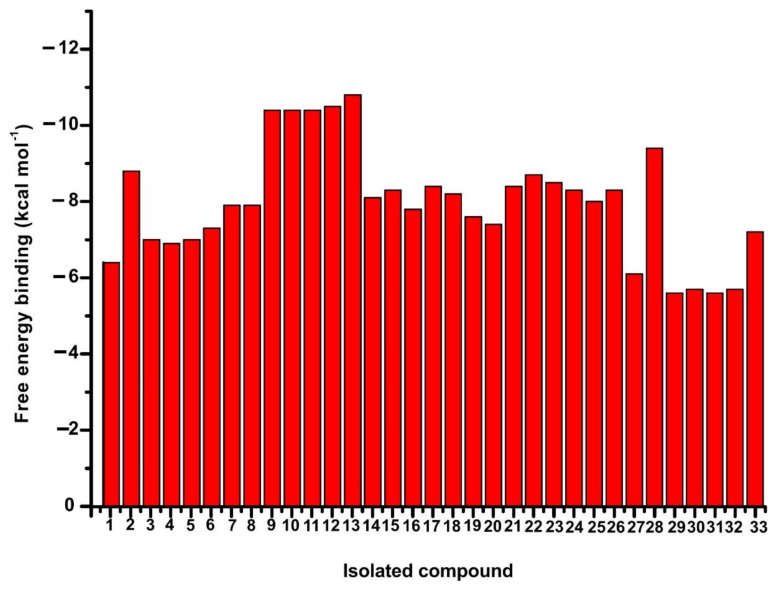
Illustration of the i*nverse virtual screening* calculated free energy binding of the 33 isolated compounds with hyaluronidase.

**Figure 5 molecules-26-03151-f005:**
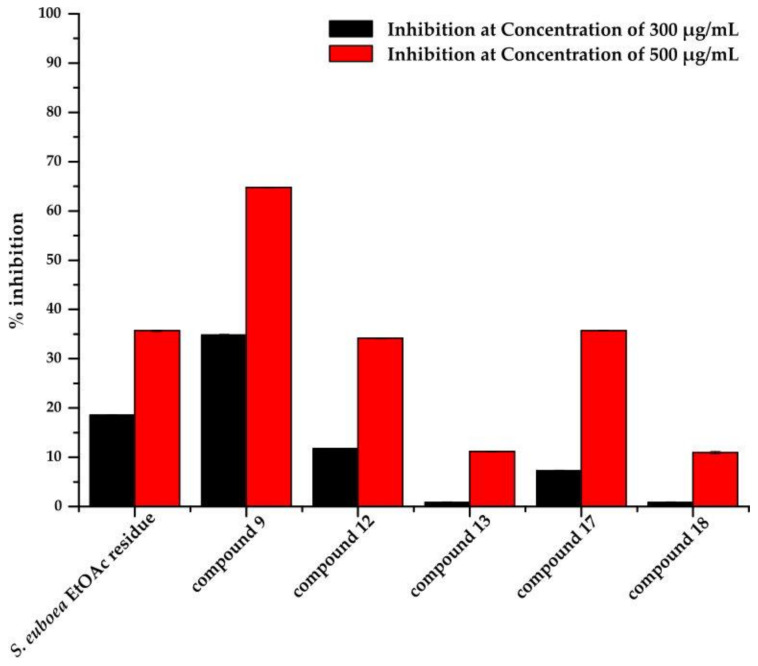
Illustration of the % inhibitory potency values of the ethyl acetate residue of *S. euboea* and the isolated compounds **9**, **12**, **13**, **17** and **18** towards hyaluronidase, at concentrations of 300 μg/mL and 500 μg/mL.

**Figure 6 molecules-26-03151-f006:**
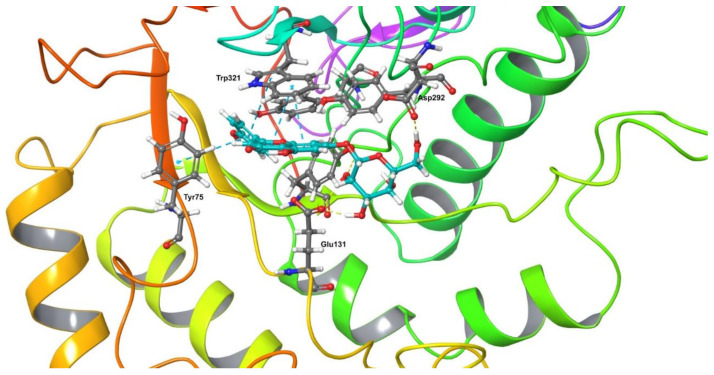
Best pose of the interaction of compound **9** (colored in light blue) with hyaluronidase. The pi-stacking interactions between the flavonoid moiety and the amino acid residues Tyr75 and Trp321 are shown with light blue dashed lines. The hydrogen bonds between the sugar moiety and the amino acid residues Asp292 and Glu131 are shown with yellow dashed lines.

**Table 1 molecules-26-03151-t001:** The main proton chemical shifts of the ^1^H-NMR spectrum of the total EtOAc residue correlated to its observed major chemical categories.

δ_H_ (ppm)	Protons	Major Chemical Categories
8.00–6.21	Aromatic	Flavonoids, Phenylethanoid glycosides, Lignans, Phenolic acids
5.88–5.70	Olefinic	Iridoids, *cis*-Phenylethanoid glycosides, Flavone *cis*-p-coumaroyl-glucosides
5.50–3.12	Methine, Methylene, Protons bonded to oxygen group	Iridoids, Lignans, Sugars
2.80	Benzylic methylene	Phenylethanoid glycosides
1.98–2.06	Methyl of acetyl groups	Iridoids, Flavonoids
1.82–0.72	Methyl	Diterpenes, Rhamnose

## Data Availability

Not applicable.

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
