# Peer review of "Anti-Ageing Potential of S. euboea Heldr. Phenolics"

_molecules, 2021, doi:10.3390/molecules26113151_

Round 1

Reviewer 1 Report

In this study, authors investigated the chemical constituents and their bioactivity against anti-ageing activity of the cultivated species of Sideritis euboea. Authors isolated 33 compounds from the S. euboea extracts using various chromatography techniques. Isolated compounds elicited using 1D and 2D-NMR techniques.  Some of compounds were reported for the first time for S. Euboea and genus Sideritis. in silico studies, isolated compounds showed different level of activity from potent to moderate activity. Authors might focus on active compounds in further for their cytotoxicity activity. The supporting material is very useful. If authors mention of ethnobotanical uses of S. euboea, would be great.  Is there any uses this species as an herbal tea? Why this species is cultivated in Greece? 

Author Response

In this study, authors investigated the chemical constituents and their bioactivity against anti-ageing activity of the cultivated species of Sideritis euboea. Authors isolated 33 compounds from the S. euboea extracts using various chromatography techniques. Isolated compounds elicited using 1D and 2D-NMR techniques.  Some of compounds were reported for the first time for S. Euboea and genus Sideritis. in silico studies, isolated compounds showed different level of activity from potent to moderate activity.

Authors might focus on active compounds in further for their cytotoxicity activity. The supporting material is very useful.

Comment #1: If authors mention of ethnobotanical uses of S. euboea, would be great.

Response: We added them in the manuscript (lines 50-52).

Comment #2:  Is there any uses this species as an herbal tea?

Response: Its infusion is used against mild gastrointestinal discomfort and common flu (lines 50-52).

Comment #3:  Why this species is cultivated in Greece?

Response: It is cultivated in order to be protected for its extinction and its overharvesting (lines 53-61). 

Reviewer 2 Report

The manuscript by Tomou et al presents a thorough and technical sound study of the chemical ingredients of the ethyl-acetate fraction of Sideritis euboea Heldr methanol extract. Overall, thirty-three compounds were isolated and identified by NMR spectroscopy, some of them for the first time in S. Euboea. Such as this manuscript deserves its publication. The authors have also evaluated the inhibitory potency of the isolated compound against hyaluronidase, a well-studied pharmaceutical target for immunodeficiency syndromes, and as a mean to facilitate tissue permeability, involved in tumor growth and angiogenesis. It is also cosmetic anti-ageing target for the development of cosmetics. Their initial screening has revealed some moderate inhibitors of the enzyme which were studied further. I would suggest measuring the IC50 (if not the Ki) values for these inhibitors instead of reporting inhibition in two different concentrations (please use molar units and not mg/mL) so it would be much easier for the reader to compare their potency with other known inhibitors of hyaluronidase. The in vitro inhibitory potency is explained satisfactory, by computational docking of the inhibitors to the 3D structure of hyaluronidase. However, their docking method is not validated (I could not find any mention of the method validation) and I would like to suggest docking first a known inhibitor of hyaluronidase and compare its docking pose with experimental evidence.

Author Response

The manuscript by Tomou et al presents a thorough and technical sound study of the chemical ingredients of the ethyl-acetate fraction of Sideritis euboea Heldr methanol extract. Overall, thirty-three compounds were isolated and identified by NMR spectroscopy, some of them for the first time in S. Euboea. Such as this manuscript deserves its publication. The authors have also evaluated the inhibitory potency of the isolated compound against hyaluronidase, a well-studied pharmaceutical target for immunodeficiency syndromes, and as a mean to facilitate tissue permeability, involved in tumor growth and angiogenesis. It is also cosmetic anti-ageing target for the development of cosmetics. Their initial screening has revealed some moderate inhibitors of the enzyme which were studied further.

Comment #1:  I would suggest measuring the IC50 (if not the Ki) values for these inhibitors instead of reporting inhibition in two different concentrations (please use molar units and not mg/mL) so it would be much easier for the reader to compare their potency with other known inhibitors of hyaluronidase.

Response: The required IC50 values of the studied compounds were calculated and expressed in μM (as also in μg/mL). Regarding the EtOAc residue (crude extract), the IC50 value was expressed in μg/mL.

Comment #2:  The in vitro inhibitory potency is explained satisfactory, by computational docking of the inhibitors to the 3D structure of hyaluronidase. However, their docking method is not validated (I could not find any mention of the method validation) and I would like to suggest docking first a known inhibitor of hyaluronidase and compare its docking pose with experimental evidence.

Response: We would like to thank the reviewer for this comment. As suggested, we have validated the docking method used in our manuscript. To validate the docking results we performed docking calculations of the well-known inhibitor of hyaluronidase liquiritigenin that its interaction with hyaluronidase has been reported (https://doi.org/10.1155/2016/9178097). For this we used a different docking software (Maestro). Both software (Maestro and Autodock) resulted on the same interaction profile of liquiritigenin with hyaluronidase (Figure S9).

Reviewer 3 Report

Tomou et al. describe the identification of 33 compounds in the extract from Sideritis euboea and the anti-hyaluronidase activity of some of them predicted by in silico screening and docking.

Minor revisions:

Give more details about the results from the inverse virtual screening.

Figure 4. Correct inverSe ...

Figure 6. Add in the text: compound 9 (in green) ...

Author Response

Tomou et al. describe the identification of 33 compounds in the extract from Sideritis euboea and the anti-hyaluronidase activity of some of them predicted by in silico screening and docking.

Minor revisions:

Comment #1:  Give more details about the results from the inverse virtual screening.

Specific results for all compounds have been reported in the revised version of our manuscript.

Comment #2: Figure 4. Correct inverSe ...

We would like to thank the reviewer for spotting this. This has been corrected in the revised version of our manuscript.

Comment #3: Figure 6. Add in the text: compound 9 (in green) ...

We would like to thank the reviewer for this comment. This has been corrected in the revised version of our manuscript and in order to make a more clearer color coding representation a color change of the ligand has been performed.

Finally, we suggest to revise our title, as follows: Anti-ageing potential of S. euboea Heldr. phenolics, due to the fact that they inhibit hyaluronidase and moreover, they possess antioxidant/anti-inflammatory activities related to the prevention of skin ageing induced by ROS.

Round 2

Reviewer 2 Report

The authors have responded satisfactory to my comments, so I recommend publication.